# Impact of the Implementation of Laboratory Information System (WWDISA) on Timely Provision of HIV-1 Viral Load Results in a Rural Area, Inhambane, Mozambique

**DOI:** 10.3390/healthcare10112167

**Published:** 2022-10-29

**Authors:** Elda Muianga Anapakala, Patrina Chongo, Isis da Barca, Tomás Dimas, Nadia Sitoe, Ruben Sebastião, Francelino Chongola, Isabel Pinto, Osvaldo Loquiha, Solon Kidane, Ilesh Jani, Sofia Viegas

**Affiliations:** 1Instituto Nacional de Saúde, Marracuene 3943, Mozambique; 2Provincial Health Directorate, Inhambane 1309, Mozambique; 3National Directorate of Medical Assistance, Ministry of Health, Maputo 264, Mozambique; 4Clinton Health Access Initiative, Maputo 1101, Mozambique; 5Departamento de Matemática e Informática, Universidade Eduardo Mondlane, Maputo 254, Mozambique; 6Association of Public Health Laboratories, Maputo 1101, Mozambique

**Keywords:** WWDISA, Mozambique, laboratory information system, implementation of laboratory information system, web-based laboratory information

## Abstract

WWDISA is an optional module of the DISA Laboratory Information system (LIS) that offers a web portal that allows access to test results over the internet for patient clinical management. This study aims to assess the applicability of using the WWDISA web application, and the lessons learned from its implementation in six health facilities in Mabote district, Inhambane province. Data from 2463 and 665 samples for HIV-viral load (HIVVL) tests, extracted from paper-based and WWDISA systems, respectively, were included, from January to December 2020. Data were simultaneously collected on a quarterly basis from both systems to allow comparison. The WWDISA turnaround time (TAT) from sample collection to results becoming available was found to be 10 (IQR: 8–12) days and significantly lower than the health unit manual logbook (*p* value < 0.001). Regarding the system efficiency, it was found that among 1978 search results, only 642 (32.5%) were found, and the main challenges according to the users were lack of connectivity (77%) and the website going down (62%). The WWDISA module has been shown to be effective in reducing the TAT, although a stable internet connection and accurate data entry are essential to make the system functional.

## 1. Introduction

A laboratory information system (LIS) is a computer software program that receives, processes, and stores data created by laboratory operations to manage information related to laboratory activities, including the management of test results. This system enables health professionals to track the progress of patients using laboratory tests [1,2,3]. Laboratory automation has enhanced the dynamics of sample testing by providing for rapid verification of results and, as a result, a reduction in turnaround time (TAT) [1,4,5].

The DISA LAB is a commercial LIS, widely used in Mozambique public laboratories, for the management of lab tests. The DISA Link System, an electronic remote test order and result receiving system, is linked to the DISA LAB LIS and serves as a sample referral system throughout the country, mainly used for HIV-1 viral load (HIVVL), early infant diagnosis of HIV, and recently for COVID-19. Another relevant DISA LAB module is the WWDISA, a web application that allows for searching for test results through a web application as soon as results are available on the central DISA LAB system. The use and application of the WWDISA has been less explored in Mozambique.

Mozambique has approximately 1748 health units and 470 labs (331 are level I labs- primary health care level) and much of the laboratory information system is paper-based. Currently, DISA LAB is used in about 25 laboratories, and DISA Link/DISA POC is used in about 290 laboratories/hubs. According to the HIVVL Implementation Guideline in Mozambique, a TAT of 15 days was stipulated, from sample reception in the laboratory to the validation of test results, with 30 days until the result is available in the patient’s clinical process [6,7]. Strategies to ensure results are available in less time, in particular in remote areas, are needed for timely patient management.

With this study, we aim to assess the applicability of the WWDISA web module and the lessons learned from its implementation for HIVVL tests, in six health facilities in Mabote District, a peripheral area in Inhambane province, from January to December 2020.

## 2. Materials and Methods

### 2.1. Study Area

The study was conducted in Mabote district, Inhambane province, where there are seven health facilities (Centro de Saúde de Mussengue, Centro de Saúde de Papatane, Centro de Saúde de Zimane, Centro de Saúde de Benzane, Centro de Saúde de Maculuve, and Centro de Saúde de Mabote), two of which have laboratories. Initially, the study comprised all seven health units. However, the Tanguane Health Unit had to be excluded from the study due to inadequate internet connectivity and a lack of electricity to charge the tablet device.

Mabote district is located in the north-east of Inhambane province (Figure 1), separated to the north from the provinces of Manica and Sofala by the River Save, to the west from Gaza province, to the east from the districts of Govuro and Inhassoro, and the south from the district of Funhalouro.

Since there is no molecular biology lab for processing HIVVL or early infant diagnosis tests in Inhambane province, all samples for those tests are sent from Inhambane to Xai-Xai Molecular Biology Laboratory in Gaza province, a nearby province. In Inhambane, the DISA Link sample referral system is used for sample recording and reporting. Samples for HIVVL are delivered with requisition forms, and data are recorded in logbooks. 

The current HIVVL sample reference system works as follows: in Mabote, samples are collected in dried blood spot (DBS) and then sent by car to the district’s main locality, Centro de Saúde de Mabote (hub), where they are recorded in the DISA Link module using the patient’s request form. The samples are then sent to the reference laboratory in Xai-Xai Provincial Hospital. As soon as the results are released, they are available through DISA Link at the hub, where can be printed and sent to other health units in the district of Mabote, where DISA Link is not available (Figure 2).

### 2.2. Sampling

Data from 2463 and 665 samples for HIV-viral load (HIVVL) tests, extracted from paper-based and WWDISA systems, respectively, were included, from January to December 2020.

### 2.3. Data Collection and Analysis

From January to December 2020, data were simultaneously collected from WWDISA and from the paper-based system, to allow comparison. Before going live, an implementation pilot was conducted (data not shown), clinicians and laboratory technicians were trained on how to use WWDISA, and the internet was provided. Data were collected quarterly to respond to the following indicators:-Time from sample collection to results availability in the system.-Time from sample collection until paper results are available and recorded in the health center.-TAT versus HIVVL.-Number of samples registered in the registration book.-Number of samples registered on DISA/Link.-Number of results searched in WWDISA.-Number of results found in WWDISA.

Evaluation visits were conducted to assess the system’s implementation and to provide assistance as needed. For discussion and experience sharing, a WhatsApp group was created. The system’s operation was remotely checked once a week or whenever there was a site complaint. A survey of 13 WWDISA users was also conducted to assess their satisfaction with the system.

### 2.4. Statistical Analysis

For the analysis comparing the median TAT between the results from the DISA (B_DISA) database and data from the logbooks (B_MABOTE), for each quarter (1, 2, 3, and 4), a Chi-squared test was used, with *p*-values plotted alongside boxplots created in R version 4.1.0 (R Foundation for Statistical Computing).

Analysis of TATs and HIVVL values was conducted by noting the difference in the median of TATs according to viral load value. Given that each sample group with different viral load values had a varied size, the Kruskal–Wallis test was used.

The change in the median of TAT, as well as how well the WWDISA functions, was measured by the number of results that were successfully researched and found to assess the impact of WWDISA. The level of user satisfaction and their reports on the challenges they experienced were also included. For inference, statistics with *p*-values < 0.05 were considered significant.

### 2.5. Ethical Considerations

Ethical approval for the study was provided by the Institutional Bioethics Committee of the Instituto Nacional de Saúde (CIBS-INS) in November 2019, with the reference number 108. Because of the nature of the study, no informed consent was required.

## 3. Results

### 3.1. Turnaround Time from Sample Collection to Results Availability

Turnaround time from sample collection to results availability is illustrated per quarter during the study period (Figure 3). In general, the TAT for the WWDISA system was found to be around 10 days (DISA_Database, in blue color) while the TAT for the health unit manual logbook (manual logbook, in yellow) varied from 25 to 55 days (*p*-value < 0.001).

### 3.2. Turnaround Time in Phases of the Referencing Chain According to the Viral Load

Turnaround time varied considerably across the reference chain and for the different viral load groups (Table 1). The median TAT from sample collection to lab registration was estimated as 10.6 (IQR: 8.4–13.3) days, and although there was no significant difference amongst the viral load groups (*p*-value = 0.2871), the highest median TAT of 11.6 (IQR: 8.5–13.4) days was observed for the viral load group of 1000–5000 cp/mL. From lab registration to analysis, the median TAT was estimated as 9.9 (IQR: 3.9–19.9) days, with the lowest TAT observed for the patients with a viral load above 5000 cp/mL (*p*-value < 0.001). From analysis to validation, the median TAT was 0.2 (IQR: 0–1.7) days, with no differences between viral load groups (*p*-value = 0.9727). The median TAT from sample collection to results availability was 10 (IQR: 8–12) days, with no significant differences amongst the viral load groups (*p*-value = 0.3041).

### 3.3. Laboratory Test Results Available in WWDISA

From January to December 2020, 2463 samples were registered in Mabote DISA Link, of which 11 were rejected due to insufficient data. When analyzing the number of searches on WWDISA, we found a total of 1978 searched results, but only 642 (32.5%) were found. Each patient’s name was searched for an average of two times before the user gave up.

### 3.4. User Challenges When Using WWDISA

During one of the monitoring visits, a survey was conducted with the participation of 13 WWDISA users. The results are shown in Figure 4. According to the findings, most users faced problems with a lack of connectivity, explained by the fact that they were in a very remote area, where the internet network is a challenge.

## 4. Discussion

This article assesses the suitability of the WWDISA module in six health units in Mabote district, Inhambane Province, a remote area in Mozambique, where access to lab results is a challenge.

Based on our analysis, in Mabote, TAT was revealed to vary greatly throughout the year using the paper-based system, while using WWDISA it showed some stability (average of 10 days). According to the HIV Viral Load Implementation Guideline in Mozambique, the maximum TAT is 30 days; however, with the paper-based system, the median TAT varied from 25 to 55 days, making patient clinical follow-up and adequate care difficult [6,8]. Many reasons can contribute to this finding, including poor patient registration and requests, incomplete registration books, unreadable writing, loss of lab test requests, delays in sample transportation to the testing site, and delays in distributing paper-based results from the hub with DISA Link to the health unit. Our findings show a substantial advantage in terms of TAT when using WWDISA, compared with the paper-based system in use in Mabote district. Other studies have shown substantial improvement when implementing informatics systems when compared with paper-based systems [8,9].

Despite this substantial advantage, there is still room for improvement from sample collection to the lab registration process (e.g., by increasing the number of days for sample transportation within the province and training the data entry staff for faster entry with quality), as this is the slowest process within the testing chain. A major concern was to find results in WWDISA. Of all the results searched, less than half were found, probably due to data entry issues. The reasons for this may be typing errors while registering samples in DISA Link, so there are no correct names in the system. The WWDISA module allows patient result searches by the patient’s first name and last name if correctly spelled. If a patient’s name is misspelled, then the sample registration code is needed to return the patient result. Other search options are needed, e.g., matching similar names, in order to solve this issue.

From a healthcare perspective, electricity is a critical parameter for delivering and improving healthcare services. Unstable electricity affects the utilization of health services [10], emergencies in medical needs [11,12], the basic functionality of healthcare facilities, and the quality, accessibility, and reliability of health services delivered [13]. In Mozambique, 19% of the total health units have no electricity [14].

Stable electricity and an internet connection are just as important to access lab results through WWDISA, since those are essential requirements to operationalize a web-based system and are possibly the reason why there were identified constraints related to the website being down. Satellite internet access might be a possible solution to be evaluated in Mabote district to make lab results accessible in a timely manner.

An information system, even if well-designed, may face several obstacles such as a lack of trained personnel, limited transportation, and large areas of coverage, and creating well-designed information systems is a difficult task that requires adequate resources, knowledge, and time to be successful [8]. The findings of this study showed that WWDISA was effective in reducing TAT, making results available for action in less time and that WWDISA’s reliability depends heavily on appropriate data entry, stable electricity, and internet connection.

It is therefore important that changes in policies and strategies in low-income countries, including Mozambique, are based on scientific evidence, and that more regionally based network-building initiatives are established to foster communities of practice and inter-institutional collaboration [15], allowing other countries already using DISALab to make use of the results from this study to define implementation strategies in their countries.

## 5. Conclusions

In this article, we discussed the implementation of a WWDISA for the management of HIVVL laboratory results in the Mabote district of Inhambane.

The WWDISA system has been shown to be effective in terms of TAT, as expected when using an electronic system. However, there are improvements needed to make the system more effective, including in the system algorithm for searching patients’ results. A LIS is not sufficient on its own, even if well-designed; it is important to have a stable internet connection, and users must be well-trained to guarantee accurate data entry.

## Figures and Tables

**Figure 1 healthcare-10-02167-f001:**
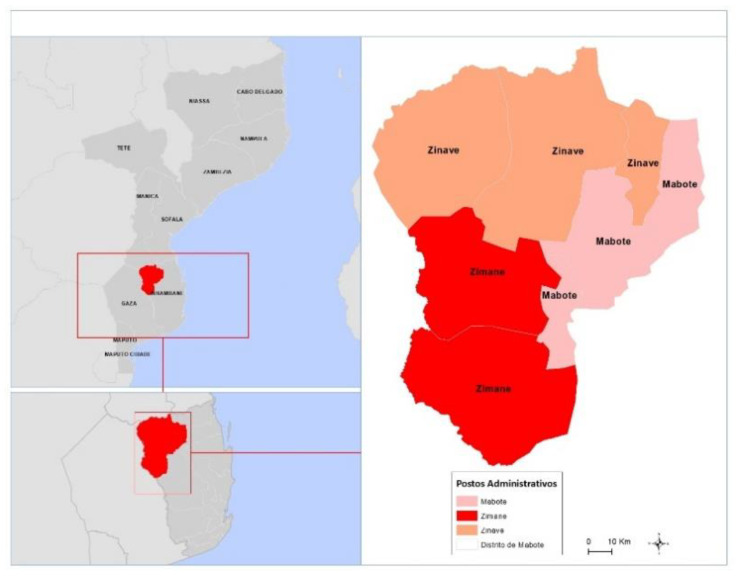
District of Mabote, Inhambane Province.

**Figure 2 healthcare-10-02167-f002:**
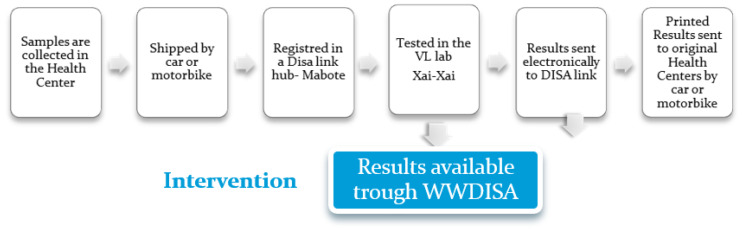
Workflow of HIVVL sample reference system.

**Figure 3 healthcare-10-02167-f003:**
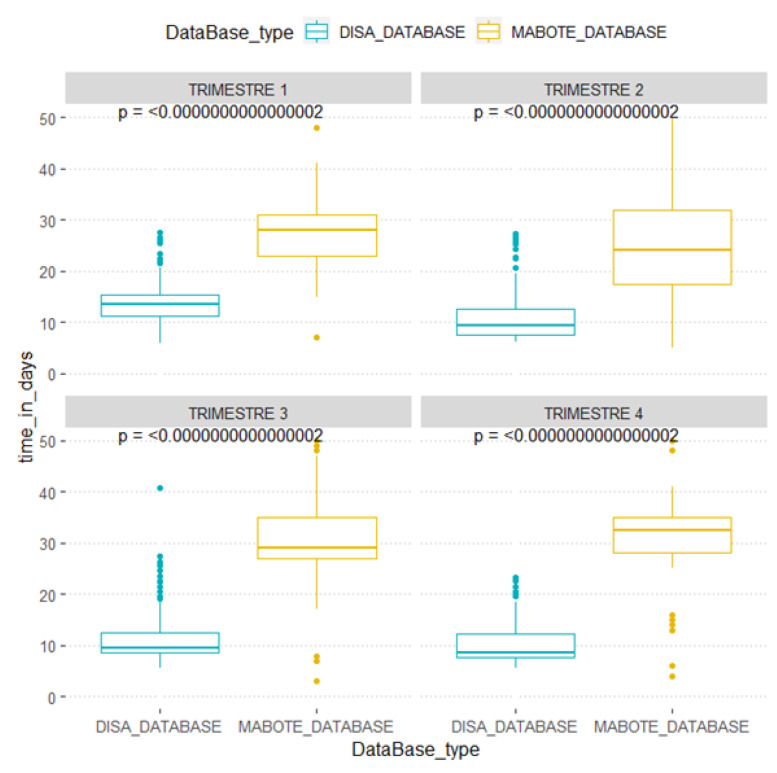
Turn the time over four quarters, from sampling to the availability of physical results, checked in laboratory journals.

**Figure 4 healthcare-10-02167-f004:**
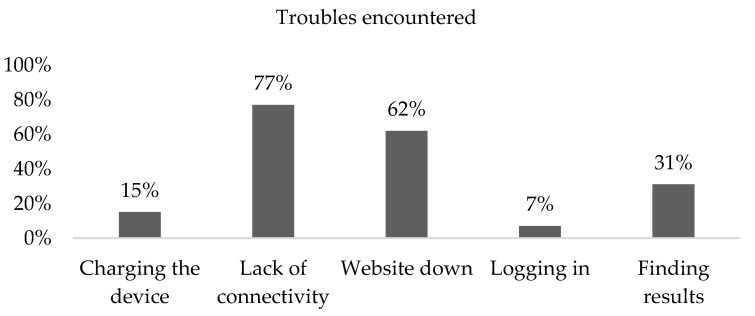
Evaluation of the user survey. The proportion of survey responses was taken once, in 13 system users.

**Table 1 healthcare-10-02167-t001:** Turn-Around Time in phases of the referencing chain according to the Viral Load.

			Sample Collecting-LAB Registration			LAB Registration-Analysis			Analysis-Validation			TAT- Sample Collection-Validation		
				IQR			IQR			IQR			IQR	
Viral Load	*n*	%	Median	Min	Max	Median	Min	Max	Median	Min	Max	Median	Min	Max
Undetectable	1765	71.7%	10.6	8.4	13.3	12.1	5.1	23.2	0.2	0.0	1.2	10	8	12
<1000 cp/mL	165	6.7%	9.6	7.6	12.6	9.9	3.0	15.8	0.2	0.1	1.9	9	8	12
1000–5000 cp/mL	197	8.0%	11.6	8.5	13.4	12.1	5.1	23.8	0.2	0.1	1.8	11	8	13
≥5000 cp/mL	216	8.8%	11.5	8.4	13.4	7.9	3.8	15.8	0.2	0.0	1.7	11	8	13
Not available	120	4.9%	8.6	7.6	12.6	0.8	0.2	1.4	11.0	11.0	11.0	9	7	12
Total	2463		10.6	8.4	13.3	9.9	3.9	19.9	0.2	0.0	1.7	10	8	12
*p*-value (Comparison of medians)			0.2871			<0.001			0.9727			0.3041		
Excluded from comparison														
IQR = Interquartile range														

## Data Availability

The data that support the findings of this study are available from the corresponding author upon reasonable request.

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
