# Peer review of "Impact of the Implementation of Laboratory Information System (WWDISA) on Timely Provision of HIV-1 Viral Load Results in a Rural Area, Inhambane, Mozambique"

_healthcare, 2022, doi:10.3390/healthcare10112167_

Round 1

Reviewer 1 Report

Although there are no major flaws in the work that is presented, I doubt whether the content is interesting enough to be presented in a scientific journal. To me this is rather an example of good management: evaluating a practice, and see where it can be improved. Moreover, the practice that is described is rather contextual. It lacks originality or real scientific value to justify sharing it with the scientific community.

Reviewer 2 Report

The paper presented here is interesting. As the authors indicate, an implementation of WWDISA for the management of HIVVL results in the Mabote district of Inhambane has been carried out. In this sense the work is in line with the objectives, the methodology and the results are interesting.

However, I would like to make some comments and suggestions to the authors in order to help the reader of the article.

The paper uses straightforward language and does not provide much information about the context in which it was developed. The lack of context makes the discussion less powerful and unclear. In fact, the authors talk about some of the problems of the health system in the area. The problem is that the reader does not have a good understanding of the medical and social reality of the region. I therefore believe that further work in this direction would be extremely beneficial.

On the other hand, the discussion is somewhat lacking. I wonder if there are other biomedical works that show, for example, whether electricity or internet connection has been important for them (Please, consider: Olayinka O. Ogunleye, Joseph O. Fadare, Jaran Eriksen, Omo Oaiya, Amos Massele, Ilse Truter, Simon J.E. Taylor, Brian Godman & Lars L. Gustafsson (2019) Reported needs of information resources, research tools, connectivity and infrastructure among African Pharmacological Scientists to improve future patient care and health, Expert Review of Clinical Pharmacology, 12:5, 481-489, DOI: 10.1080/17512433.2019.1605903). I also wonder if there are any strategies that are trying to solve this problem and whether it affects other techniques.

The authors indicate " there is still room for improvement of the lab registration process as shown by the results, from sample collection to lab registration being the slowest process within the testing chain" (lines 174 and 175), I humbly believe that it would be important for them to make an effort to explain the mechanisms of improvement. Precisely, I think it would help a lot if the authors would make a flow chart of the process for sample collection and subsequent analysis.

Pan et al (2021) (Pan, J., Zhong, Y., Young, S. et al. Collaboration on evidence synthesis in Africa: a network study of growing research capacity. Health Res Policy Sys 19, 126 (2021). https://doi.org/10.1186/s12961-021-00774-2) talk about the interaction between the health system and social institutions. In this sense, I wondered if there would be any possibility to theorise the use of their strategy as a possible model to operationalise a strategy of analysis. I also wondered whether it might be possible for social institutions to take part in it. Furthermore, I also wonder if this strategy could be networked and, in the near future, incorporate new actors.

In short, I think the work lacks a bit of context, analysis and depth. It is undoubtedly interesting research, but it needs more effort to achieve the quality the work deserves.

Reviewer 3 Report

Brief Summary

Anapakala et al. embark on a journey to improve access of clinical data to both patients and clinicians. Specifically, they focus on implementing an electronic laboratory information system to provide the test results for HIV viral load. Furthermore, the new system improved turnaround times from sample collection to results by more than twice. 

Significance

Decades of research has turned HIV from a deadly disease to a chronic lifelong infection. Fast and reliable access to patient medical information is key to successful managing of HIV infection. Of special interest is the HIV viral load of a patient, critical for evaluating treatment impact and essential for changing to new course of treatment if needed. Furthermore, assessing the implementation of new electronic systems is especially important in remote areas facing unique challenges, such as difficulty accessing lab results and lack of connectivity. 

Recommendations: 

I recommend accepting this paper with minor revisions for publication at the Healthcare Journal. I am listing below minor suggestions for clarifying details described in this commentary. 

Comments: 

Line 10: “quoter basis”, please check if you meant at “quarter basis”. 

Figure 1: Resolution of figure 1 was low in the file I received. Please make sure the text in the figure is readable in the final version. 

Figure 2 title: “turn the time”, please change to “turn around time”.

Table 1: Please mention that time is measured in days in this table. 

Discussion: Your results found that electricity and internet connection pose a challenge to access lab results through WWDISA. If possible, could you please elaborate on the reliability of the previous system? Specifically, did the paper-based system face less or more problems with finding the patient data? 

Round 2

Reviewer 2 Report

The modifications that have been made to the paper do it clearer and more understandable. The work was interesting in its first version. In this second version, the depth of the tool evaluated, as well as the discussion, has been improved.